# HYPED: A Multimodal HYbrid Perturbation Gene Expression and Imaging Dataset

## Abstract

Integrating multimodal, high-resolution biological data is a useful way to characterize biological processes, such as how cells respond to perturbations. Cell perturbation prediction is a major experimental challenge and has motivated substantial research in machine learning for biology. In this work, we generated a multimodal benchmark dataset that captures the dynamic response of human fibroblasts to transient transcription factor perturbations. We performed time-series live cell imaging with fluorescent cell cycle reporters over 72 hours and collected long-read single-cell RNA sequencing data from the same population of cells. We release the processed dataset, preprocessing pipelines and benchmarking code along with the evaluation of existing models using our data as ground truth. This work supports the development and evaluation of machine learning methods for modeling dynamical systems from multimodal datasets. HYPED consists of RNA sequencing data from approximately 20,000 cells and 203 imaging timepoints across four experimental conditions, totaling 2030 imaging frames. HYPED makes the cell perturbation problem accessible to machine learning researchers with state-of-the-art experimental data.

## 1 Introduction

Direct cell reprogramming, or the process of converting one cell type into another without passing through an intermediate pluripotent state, holds promise for personalized and regenerative medicine (65; 53). Reprogramming requires the introduction of targeted perturbations capable of altering cell state, most often through the introduction of cell-type specific transcription factors (TFs) (22; 61). Current methods for delivering TFs, such as lentiviral vectors or CRISPR-based systems, permanently alter the cell's DNA. This irreversible modification raises concerns about uncharacterized and potentially harmful effects, limiting their utility for a wide range of research and clinical applications. To address this limitation, TF delivery via transient RNA-based methods have emerged as safer alternatives (57; 23).

Despite recent progress in the field of direct reprogramming, key challenges remain: conversion efficiency remains low and identifying new TF combinations is difficult due to limited knowledge of their combinatorial effects (31; 61). Evaluating the effects of perturbations is essential for optimizing existing cell reprogramming protocols. However, most existing datasets are limited to a single data modality and do not use transient TF delivery methods (16; 49; 70; 20; 13; 4).

This lack of adequate data poses a challenge for evaluating and benchmarking machine learning models for cell perturbation prediction. In this work, we collected high-resolution, long-read single-cell transcriptomic data and live-cell fluorescent microscopy data from a state-of-the-art TF reprogramming protocol on human fibroblasts using a combination of transient forced overexpression (*MYOD1*) and gene suppression (*PRRX1*). Our dataset is a valuable resource for the machine-learning evaluation of cell perturbations.

**Biological Background.** The cell reprogramming problem originated in the 1980s with H. Weintraub's use of the TF MyoD to convert fibroblasts into skeletal muscle (66). In the early 2000s, S. Yamanaka reprogrammed the first induced pluripotent stem cells (iPSC) (53). Both approaches typically use viral vectors to permanently integrate modified genes into the host cell genome. Many contemporary perturbation screens use a CRISPR-based approach (e.g., Perturb-seq) to modulate

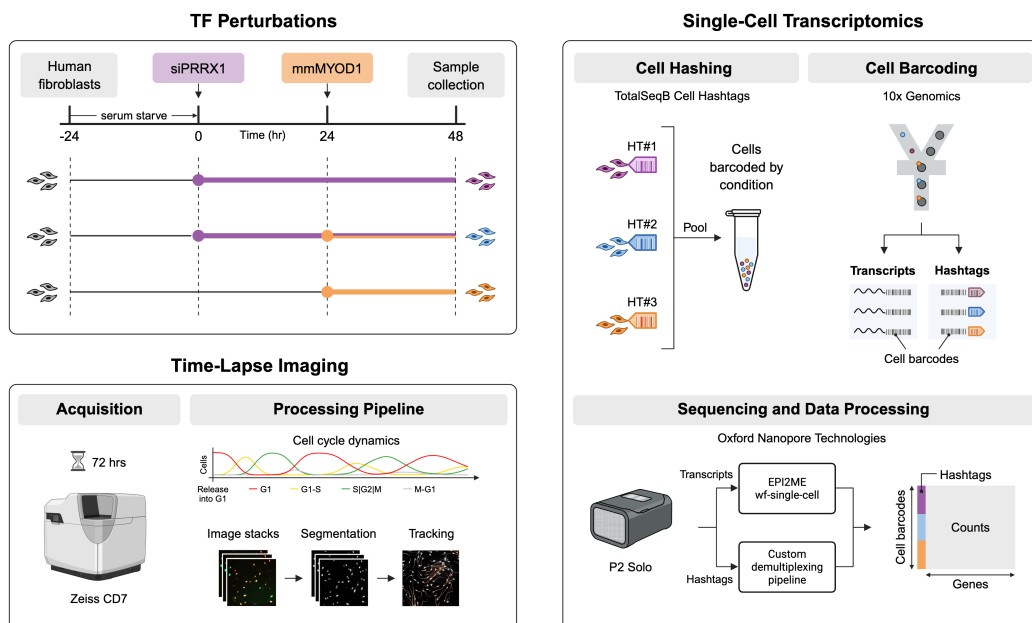

Figure 1: **Experimental Dataset.** The HYPED dataset includes time-series imaging and long-read sequencing to characterize transient cell perturbations. **Top left:** Human fibroblasts are perturbed using combinations of *MYOD1* overexpression and *PRRX1* attenuation, delivered via modified mRNA and siRNA, respectively. **Bottom left:** Live-cell fluorescent imaging with the Zeiss Celldiscoverer 7 (CD7) captures the temporal dynamics of the cell cycle under TF perturbation. **Right:** Long-read single-cell sequencing captures the final transcriptional state of the perturbed cells.

gene expression, but also irreversibly modify the host cell genome (10). These methods have produced useful datasets for perturbation prediction using machine learning. However, genome modification itself can result in off-target effects, making non-integrating approaches desirable from a safety and quality standpoint.

Non-integrating methods including modified mRNA (mmRNA) and small interfering RNA (siRNA) have been successfully used for direct reprogramming (62; 60; 56). mmRNA enables transient, high-level expression of reprogramming TFs without genomic integration, while siRNA facilitates the attenuation of endogenous mRNAs that maintain the original cell identity. Together, these approaches promote efficient and controlled lineage conversion without permanent genetic alteration (50; 34). To our knowledge, we present the first cell perturbation dataset based on state-of-the-art experimental methods for machine learning.

**Data Modalities.** HYPED is a multimodal dataset that combines high-throughput sequencing with live-cell imaging (fig. 1). Recent advancements in long-read omics technologies, such as single-cell RNA sequencing (scRNA-seq), have expanded our ability to investigate complex intracellular mechanisms across multiple regulatory levels (26; 58; 25). In particular, scRNA-seq with Oxford Nanopore Technologies enables the direct sequencing of full-length RNA transcripts, providing insights into transcriptional regulation and cell state dynamics. The long-read capability of Nanopore sequencing allows for the identification of alternative splicing events, quantification of isoform diversity, and detection of RNA modifications, which are often inaccessible features through short-read platforms (24; 26). This enables a more accurate and comprehensive characterization of the transcriptome, facilitating the study of gene expression heterogeneity at the single-cell level.

RNA sequencing remains one of the most widely used data modalities to assess cell responses to perturbation (43; 55; 45). However, sequencing requires cell lysis, resulting in destruction of the cell during sample preparation. In contrast, advances in live-cell imaging technologies have significantly improved spatial resolution and throughput, offering a non destructive, time-resolved method for

Table 1: **Overview of Related Single Cell Datasets.** Overview of benchmark datasets used in machine learning for cell imaging, sequencing, and perturbation.

| Dataset ML Task | | Imaging | Sequencing | Perturbation |
|---|---|---|---|---|
| *Only Imaging* | | | | |
| LIVECell | (11) | ✓ | | |
| NeurIPS Competition 2022 | (33) | ✓ | | |
| *Only Sequencing* | | | | |
| CellXGene | (41) | | ✓ | |
| NeurIPS Competition 2021 | (32) | | ✓ | |
| *Multi-Modal Perturbation (Perturbation + Sequencing)* | | | | |
| scPerturb | (18) | | ✓ | ✓ |
| PerturbBase | (64) | | ✓ | ✓ |
| sc-pert | (27) | | ✓ | ✓ |
| NeurIPS Competition 2023 | (7) | | ✓ | ✓ |
| *Multi-Modal (Imaging + Perturbation + Sequencing)* | | | | |
| **HYPED** (ours) | | ✓ | ✓ | ✓ |

studying dynamic cell processes. These systems enable high-resolution monitoring of features like cell morphology, proliferation, migration, and apoptosis over time. When combined with fluorescent reporters such as the Fluorescent Ubiquitination-based Cell Cycle Indicator (FUCCI) system (17), live-cell imaging allows real-time tracking of specific protein expression to explore intracellular drivers of phenotypic changes (48). FUCCI distinguishes cell cycle phases by leveraging fluorescently-tagged proteins whose degradation is regulated by the cell cycle. Specifically, the Incucyte Cell Cycle Reporter (Sartorius 4779), uses the FUCCI system to capture cell cycle dependent changes in the expression patterns of Geminin and Cdt1 by linking fluorescent proteins TagGFP2 (Green) and mKate2 (Red) that can then be monitored with live-cell imaging.

This approach provides a dynamic view of cell cycle progression in live cells (40) that are not accounted for in existing machine learning perturbation models. In the HYPED dataset, time-series imaging offers high temporal resolution to capture perturbation dynamics, complementing sequencing-based approaches. This addition introduces a data modality for machine learning studies of cell perturbations, enabling multimodal and temporal analysis of the perturbation prediction problem (see table 1).

**Machine Learning Approaches.** Given the significance of the cell reprogramming problem, a wide range of machine learning approaches have been developed. Early efforts include graph- and network-based methods (42; 12), as well as studies that frame reprogramming through system identification, trajectory optimization, and control theory (43; 6). More recent work leverages transformers and other modern architectures to represent and embed cell states measured with sequencing (55; 38; 9). Although these approaches have achieved successes at the machine learning problem, there remain substantial gaps in the success of experimental perturbations, which motivates the development and application of new models to state-of-the-art experimental datasets.

**Contributions.** Imaging and transcriptomics each offer distinct yet complementary insights into cell biology; imaging captures spatial and morphological dynamics, while transcriptomics reveals the molecular and regulatory state of the cell. However, these modalities are typically collected in isolation, providing only fragmented views of genetic perturbations. Integrating these experimental modalities promises a more comprehensive understanding of the cell as a dynamical system, revealing how gene expression and physical behavior co-evolve over time. This integration is technically challenging, as it involves harmonizing data with vastly different structures, resolutions, and temporal characteristics. Emerging machine learning approaches, particularly multimodal representation learning and generative modeling, offer a powerful bridge, enabling alignment, translation, and predictive modeling across these disparate data types. Leveraging these computational frameworks

is key to unlocking unified models of cell behavior that incorporate both molecular and phenotypic dimensions.

To address this challenge, we designed and performed a perturbation experiment that captured live-cell imaging and scRNA-seq from the same initial population of cells that were treated under identical conditions and cell cycle synchronized, providing a state-of-the-art ground truth dataset for a variety of challenges, including improving perturbation prediction outcomes. We overcome several limitations of existing datasets, which lack synchronized, time-resolved measurements across modalities. Our contributions are a valuable resource for benchmarking multimodal models as well as a framework for studying dynamic cell processes through an integrative lens.

## 2 RELATED WORK

### 2.1 CELL REPROGRAMMING

Direct cell reprogramming is a complex process that involves activating lineage-specific target networks while simultaneously silencing the original cell identity (36). Most reprogramming strategies rely on viral vectors or plasmids to deliver lineage-specifying genes. Pioneer factors, such as *MYOD1* in muscle (66) or *OCT4* in pluripotency (53), can initiate reprogramming but often require cooperative TFs to fully activate and maintain the target gene regulatory network (19). Without the proper TFs, most cells fail to convert fully, instead stalling in unstable intermediate states (8; 37). Researchers have mapped regulatory networks that govern cell identity and have developed libraries of transcription factors capable of inducing cell transitions (47; 69; 67; 28).

In parallel, RNA-based approaches have gained traction as non-integrative tools for cell reprogramming (63; 62; 60). Synthetic mmRNA delivery allows for transient and tunable expression of reprogramming factors, minimizing risk of genomic integration and long-term mutagenesis (50; 6; 5). Similarly, siRNA can be used to transiently suppress endogenous factors that stabilize native cell identity (56; 30). While RNA-based systems offer less fine-tuned control compared to viral or plasmid-based delivery methods, they have significantly greater clinical promise (23). Their transient nature is well suited to control the cell cycle and reduces the risk of mutagenesis and tumorigenesis, making them attractive for therapeutic applications (3). As RNA delivery technologies and engineering strategies mature, RNA-based reprogramming offers a safe path to clinical translation without compromising efficacy (31).

### 2.2 DATASETS AND MODELS FOR GENE PERTURBATIONS

Gene perturbations are a core laboratory technique for cell reprogramming and have become a focus of machine learning in biology. RNA-seq is the primary technique for measuring cell responses to perturbations, with scRNA-seq enabling high-resolution analysis of development, disease, and treatment effects. This has led to the development of large single cell atlases (21; 52; 14; 54) and foundation models (55; 15; 51). Together, these datasets and models provide a rich resource for developing machine learning strategies to infer cell identity and predict perturbation outcomes.

Perturb-seq, which combines CRISPR perturbations with scRNA-seq, enables high-throughput measurement of gene function and has led to the creation of several public datasets and resources (10). The Gene Perturbation Atlas (GPA) compiles single-gene perturbation data across diverse cell types to systematically assess how individual genes influence cell identity (68). The Perturbation Cell and Tissue Atlas (PCTA) extends this effort by integrating genetic and environmental perturbations with molecular and imaging readouts to support causal inference in cell and tissue biology (44).

Early computational frameworks for direct reprogramming combine gene expression, regulatory networks, and epigenetic information to predict optimal TF combinations (29; 43; 42; 12). However, the low efficiency of current reprogramming protocols has positioned cell perturbation prediction as a standard machine learning challenge (45; 55; 27; 72). While many models focus on forward prediction—forecasting the cell response to a given perturbation—few address how to identify the optimal perturbation strategy. Moreover, existing methods often overlook important biological priors such as cell dynamics, cell cycle phase, and other factors known to influence reprogramming success (71).

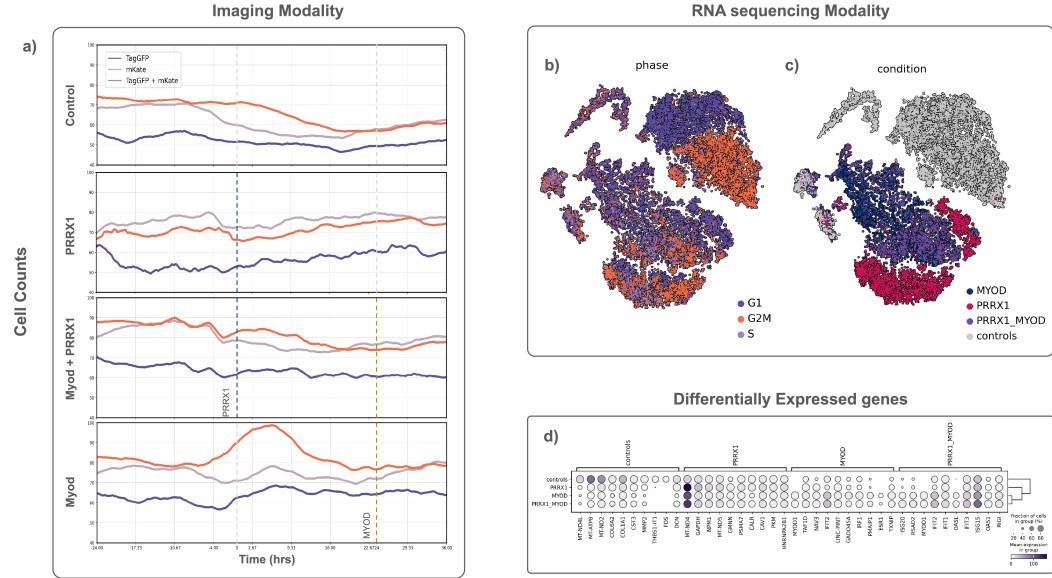

Figure 2: **HYPED Dataset Overview. (a)** Shows the Distribution of cells across different cell cycle phases, quantified from Incucyte imaging data across the different fluorescence channels. The **(b)** t-SNE projection of cells colored by cell cycle phase shows **(c)** t-SNE projection of cells colored by experimental condition. **(d)** Top 10 differentially expressed genes across experimental conditions.

## 3 DATASET

We present a multimodal dataset that captures the dynamic response of human fibroblasts to transcription factor (TF) perturbations. The dataset includes long-read scRNA-seq and time-lapse imaging data collected under four experimental conditions: *MYOD1* activation via modified mRNA (mmMYOD1), *PRRX1* suppression via siRNA (siPRRX1), sequential dual perturbation (siPRRX1 followed by mm-MYOD1), and an unperturbed control. The scRNA-seq dataset comprises approximately 20,000 cells (∼11,000 perturbed, ∼9,000 controls), enabling isoform-level transcriptomic profiling. Time-lapse imaging data were collected at 20 minute intervals over 72 hours, capturing high-resolution trajectories of cell cycle dynamics and morphological changes. Sequencing data contains of 3 replicates in separate wells, that were pooled together for single-cell barcoding and sequencing. For the Imaging data, the negative controls and the MYOD + PRRX1 condition each contain three replicates, while the MYOD and PRRX1 conditions contain two replicates each. All replicates are included within the released dataset.

### 3.1 EXPERIMENTAL SETUP

Human neonatal foreskin fibroblasts (BJ, ATCC CRL-2522) carrying the Incucyte Cell Cycle Reporter (Sartorius 4779) were used for perturbation experiments. Cells were cultured at $37°C$ in 5% $CO_2$ on standard cultureware in full media (FM; Dulbecco's Modified Eagle's Medium (DMEM, Gibco 11965-092) + 10% Fetal Bovine Serum (FBS, Corning 35-015-CV) + 1% Penicillin-Streptomycin (P/S, Gibco 15140-122)). Cells were seeded at 1.03 - 1.33 x $10^4$ cells/cm$^2$ in 6- and 48-well plates, followed by 24 hours of serum starvation (0.2% FBS) for G0/G1 phase synchronization. At t=0h, cells were released from serum starvation, and respective cells were transfected with PRRX1 siRNA (25 nM; Dharmacon) in FM using the TransIT-X2 Dynamic Delivery System (Mirus MIR6000). At t=24h, siRNA was washed out with FM. Respective cells were then transfected with MYOD1 modified mRNA (1 ng/$\mu$l; Pseudouridine + Silica purification, Trilink WOTL39876) in FM supplemented with 2 $\mu$M 4OH-Tamoxifen (Sigma-Aldrich H7904) using the Lipofectamine MessengerMAX Transfection Reagent (Invitrogen 100026485). At t=48h, cells were harvested with TrypLE Express (Gibco 12604-013) for scRNA-seq sample preparation. For imaging plates, cells were treated the

Table 2: **Imaging Data Structure.** These dimensions are used to operate on the imaging files, named according to CZI specification.

| Symbol | Description |
|--------|-------------|
| H | Phase |
| S | Scene (plate/well) |
| T | Time |
| C | Channel |
| Z | Depth dimension |
| M | Mosaic (determines sub-tile) |
| Y | Vertical axis |
| X | Horizontal axis |

same as described above with two differences: (1) Cells were cultured in imaging media (FluoroBrite DMEM (Gibco A18967-01) + 10% FBS + 1% P/S) instead of standard FM; (2) Cells were stained with 250 nM SiR-DNA (Cytoskeleton, Inc. CY-SC007) at t=-24h to enable tracking of cell nuclei.

### 3.2 SINGLE CELL RNA SEQUENCING WITH LONG READS

Cells from siPRRX1, siPRRX1/mmMYOD1, and mmMYOD1 were individually labeled with TotalSeq-B human cell hashing antibodies (BioLegend, Cat# 394631, #394633, and #394635, respectively). 20,000 cells from each condition were mixed together (viability >98%) and single cell barcoded on the 10x Genomics Chromium Controller (X) using the Next GEM Single Cell 3' Kit V4. Barcoded cDNA amplicons from transcript and hashing libraries were prepared according to the Oxford Nanopore Technologies (ONT) 3' cDNA protocol (SQK-LSK114, SST_9198_v114_revJ_13Nov2024). Libraries were assessed for quality following ONT recommendations. Prepared libraries were sequenced on the ONT PromethION Solo 2 (P2) sequencer. Raw reads were base-called with Dorado v0.9.1 (2) using the High Accuracy basecalling model and stored as fastq files.

For unperturbed controls, fibroblasts were sorted by fluorescence activated cell sorting (FACS) into their respective cell cycle phases after staining with 16 $\mu$M Hoechst 33342 for 50 minutes. Cells in G1, S, and G2/M were individually labeled with TotalSeq-B hashing antibodies (BioLegend, Cat# 394631, #394603, #394605, respectively), mixed, and single cell barcoded (viability >97%). Transcript and hashing libraries were processed and sequenced as described above.

Raw sequencing files obtained from scRNA-seq of the perturbed cells and controls were processed using the EPI2ME wf-single-cell pipeline v3.1.0 (1) with the human GRCh38 reference genome. The processed gene expression matrices were combined into a single AnnData object containing cells as rows (obs) and genes as columns (var) (59).

Respective perturbation conditions (siPRRX1, siPRRX1/mmMYOD1, mmMYOD1) were assigned to individual cell barcodes using a custom demultiplexing pipeline. We add the $assigned\_condition$ column to AnnData.obs.

### 3.3 TIME SERIES MICROSCOPY

The Zeiss Celldiscoverer 7 (CD7) live-cell imaging system was used to capture time-lapse images over the course of perturbation experiments. Oblique contrast and fluorescence microscopy was performed with a Plan-Apochromat 20x/0.7 objective and 0.5x tube lens. Images were taken using an Axiocam 506 with 14 bit resolution. Cells were imaged at 37°C in 5% $CO_2$ in imaging media. Images were captured every 20 minutes over 72 hours. Raw CZI files were exported from the Zen Blue 3.0 software for image processing and downstream analyses.

## 3.4 DATA STORAGE

The HYPED dataset is made available on kaggle along with the software used to process the raw experimental data from the sequencers. Each modalitiy of the dataset is structured separately. The raw CZI files exported from the microscope contain data with dimensions HSTCZMYX (table 2). The four imaging channels correspond to markers that can be used to delineate the cell cycle phase:

- Cy5 - cell nuclei
- MKate - G2/M phase
- tagGFP - G1 phase
- Oblique - oblique contrast

Each scene is associated with one of the four experimental conditions. We extract only the relevant scenes across time frames where each frame is split into mosaics of $6 \times 5$. These mosaic tiles are stitched together to get the full frame. Each frame was then converted into a four-mode numpy array (T, C, X, Y) with dimensions (202, 4, 2826, 3245). To make the dataset standardized and accessible, tensor data of each frame was embedded into a HDF5 (Hierarchical Data Format). Each HDF5 file consists of a single frame with the four channels at a particular image capture point of the experiment. The dataset is designed and to load frames across any range of time points into compute memory

The long read sequencing gene expression data is stored in the HDF5 AnnData format *.h5ad*, a commonly-used HDF5-based format with extensive support in Python and R. This format stores the measured gene expression in each single cell in a cell by gene matrix, with the rows and columns annotated to correspond to the genes and experimental conditions of the cells.

## 4 BENCHMARKING

**Sequencing.** We compare the ground truth results obtained from the experiment and the outputs from predictive models. We use two models to benchmark: the foundation model Geneformer (55) and the perturbation prediction model GEARS (46) and report on cosine similarity, Mean Squared Error (MSE), Mean Absolute Error (MAE) and Pearson correlation coefficient between the predicted and observed values (see table 3).

Geneformer is a Transformer-based Deep Learning model which utilizes unsupervised self-attention mechanism to generate Transcriptomic representations. This generates embeddings in a lower-dimensional space that can be fine-tuned for a variety of tasks.

We tokenize the transcriptome of each cell in the data set with rank value encoding. For the in-silico perturbation, we set the expression level of *MYOD1* to a value greater than the highest expressed gene, which places it at the top of the ranked value encoding, simulating the activation of *MYOD1* within the cell. Similarly, we simulate suppression of *PRRX1* by setting the expression level to zero before rank value encoding. The tokenized data is then run through Geneformer generating embeddings. This embedded data is compared with those of experimentally perturbed cells, and the metrics are reported.

GEARS is a Deep Learning perturbation prediction model which can predict gene expression vectors under certain perturbation conditions. The GEARS model takes in a combinations of perturbed genes as input and predicts output gene-expression vectors. The model architecture supports splitting the dataset based on different perturbation combinations. The model is first fine tuned with existing perturbation data (39) which was split into

- Train - 139 combinations
- Test - 107 combinations
- Validation - 31 combinations

This pretrained model is further fine tuned with the control group data from our experiment as 3 new conditions. This is the suggested method from the authors of GEARS to integrate new datasets into the model.

We then performed in silico prediction for all three conditions using GEARS. This generates the predicted gene expression vector for the top 500 genes which we compare with the average expression of our held out conditional cells, and calculate different metrics.

---

**Algorithm 1** Geneformer Validation Experiment

---

**Require:** perturbed_group, control_group, perturbation_genes
**Ensure:** cos_sim
 1: // Initialize lists to store control and perturbed group embeddings
 2: $E_c, E_p \leftarrow [\,], [\,]$
 3:
 4: // Compute Geneformer embeddings of perturbed data
 5: **for** each cell $x$ in perturbed_group **do**
 6:    $e \leftarrow$ Geneformer$(x)$
 7:    $E_p \leftarrow [E_p\ e]$
 8: **end for**
 9:
10: // Compute Geneformer embeddings with in silico perturbations
11: **for** each cell $x$ in control_group **do**
12:    **if** MYOD $\in$ perturbation_genes **then**
12:      $x[\text{MYOD}] \leftarrow$ max$(x)$
13:    **end if**
14:    **if** PRRX1 $\in$ perturbation_genes **then**
14:      $x[\text{PRRX1}] \leftarrow 0$
15:    **end if**
16:    $e \leftarrow$ Geneformer$(x)$
17:    $E_c \leftarrow [E_c\ e]$
18: **end for**
19:
20: // Determine the average embedding of each experimental group
21: $e_p \leftarrow$ mean$(E_p)$
22: $e_c \leftarrow$ mean$(E_c)$
23:
24: **Return** CosineSimilarity$(u, v)$ =0

---

Table 3: **Benchmarking Perturbation Models Against Experimental Data.** Single-cell control data were computationally perturbed using Geneformer and GEARS, and the resulting profiles were compared to experimentally perturbed counterparts.

| Model | Perturbation | Cell Cycle | Cosine Sim. | MSE | MAE | Pearson |
|---|---|---|---|---|---|---|
| Geneformer | + MYOD1 | G1 | 0.9808 | 0.0148 | 0.0902 | 0.9807 |
| | | S | 0.9776 | 0.0172 | 0.0988 | 0.9775 |
| | | G2/M | 0.9715 | 0.0221 | 0.1119 | 0.9714 |
| | − PRRX1 | G1 | 0.9913 | 0.0068 | 0.0605 | 0.9913 |
| | | S | 0.9900 | 0.0078 | 0.0660 | 0.9900 |
| | | G2/M | 0.9900 | 0.0078 | 0.0680 | 0.9900 |
| | − PRRX1 + MYOD1 | G1 | 0.9902 | 0.0076 | 0.0645 | 0.9902 |
| | | S | 0.9887 | 0.0086 | 0.0697 | 0.9887 |
| | | G2/M | 0.9851 | 0.0115 | 0.0810 | 0.9851 |
| GEARS | − PRRX1 | All Phases | 0.9518 | 0.0502 | 0.1079 | 0.9358 |
| | + MYOD1 | All Phases | 0.9741 | 0.0129 | 0.0622 | 0.9625 |
| | − PRRX1 + MYOD1 | All Phases | 0.9772 | 0.0116 | 0.0598 | 0.9676 |

The high cosine similarities observed for both the GEARS and Geneformer predicted gene expression indicate that the trained machine learning models reflect experimental behaviour. HYPED dataset provides a strong empirical ground truth for training and evaluating future foundation models that can leverage both modalities.

Table 4: **Image Benchmarking.** Image Quality Metrics Averaged Across Timepoints.

| Model | MSE | PSNR (dB) | SSIM | Calculated SNR | Denoised SNR |
|-------|-----|-----------|------|----------------|--------------|
| FM2S | 0.000146 | 86.480553 | 0.877678 | 14.082471 | 15.051044 |

**Imaging.**  To evaluate the performance of noise removal techniques, we performed a benchmarking task on our imaging dataset. The extraction of regions of interest was performed through a standard pipeline using the following steps:

1. **Clipping and normalization:** For each channel, intensities above 98% were clipped and scaled to [0,1] range.

2. **Channel stacking:** mKate and TagGFP channels were stacked on each other to get the G1-S phase transitions.

3. **Gaussian filtering and Otsu thresholding:**  Used to separate foreground cells from background.

4. **Watershed Algorithm:** Was used to differentiate individual cell boundaries.

5. **Segment mapping:** Segmented regions were converted to binary masks

6. **Tracking:** Bayesian tracking was used to assign object identnties over time.

These cells are then labeled as foreground, and everything else is considered noise and the Signal-to-Noise Ratio is calculated. We then run the images through a machine learning algorithm and PSNR and SSIM were calculated. FM2S (35) is a Deep Neural Network based Fluorescent Microscopy Imaging denoiser. We Run this model on different conditions of our dataset and report the average metrics (see table 4).

## 5 DISCUSSION

Here, we introduced the HYPED benchmark dataset for cell reprogramming, designed using state-of-the-art perturbation techniques. Unlike widely used lentiviral, CRISPR, and Perturb-seq datasets which can have unforeseen off-target effects, HYPED offers the first multimodal perturbation dataset generated using transient RNA application for improved learning and prediction of perturbation dynamics.

Moreover, HYPED's live cell imaging provides one of the first perturbation datasets to capture cell cycle dynamics with perturbations. It has long been recognized hat cell perturbations are transient processes, yet many current models including GEARS, Geneformer, scGPT, and others do not account for or model the dynamics. This dataset provides an improved opportunity to train models that account for perturbation and biological dynamics at a higher temporal resolution and under current experimental conditions than previously possible.

To advance cell reprogramming and related biological problems, it is essential that machine learning models are aligned with the capabilities and limitations of contemporary experiments by training on modern experimental data modalities. Many existing approaches are trained on idealized or outdated datasets that fail to reflect the transient, dynamic nature of real biological systems. By employing modified mmRNA and siRNA and including live-cell imaging with long-read single-cell transcriptomics, the HYPED dataset serves as a resource for researchers to develop models that are not only more predictive, but also more actionable in laboratory and clinical settings.

## 6 LIMITATIONS

We identified two main limitations in our study. First, the experiment conducted includes the effects of three unique transcription factor perturbations captured on skin fibroblasts. Although this describes the activity of a small subset, the vast combinatorial space of transcription factors and cell types remains largely unexplored. Second, we considered a limited number of tasks for each modality, but there are a variety of challenges that can be explored with our dataset.

## 7 ETHICAL CONSIDERATIONS

All resources provided as part of this cell reprogramming study are strictly for research purposes only and should not be used in clinical settings and diagnostic procedures. No sensitive information is included in the dataset. With the aforementioned restrictions, we have not identified any potential adverse impacts from the HYPED dataset.

### ACKNOWLEDGMENTS

We would like to thank all Rajapakse lab members for helpful and inspiring discussions. This work was supported by the Defense Advanced Research Projects Agency (DARPA) award number [HR00112490472 to I.R.], the Air Force Office of Scientific Research (AFOSR) award number [FA9550-22-1-0215 to I.R.], support from NVIDIA [to I.R.], and support from the National Institute of General Medical Sciences (NIGMS) award number [GM150581 to J.P.].

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
