# OpenReview forum: "HYPED: A Multimodal HYbrid Perturbation Gene Expression and Imaging Dataset"
_ICLR.cc/2026/Conference — ICLR 2026 Conference Desk Rejected Submission_

### Official Review · Reviewer_G18c · 2025-10-30

**Soundness:** 3
**Presentation:** 3
**Contribution:** 3
**Rating:** 6
**Confidence:** 4

**Summary:**

This paper presents HYPED, a multimodal benchmark dataset for cellular reprogramming that integrates time-series live-cell imaging with long-read single-cell RNA sequencing (scRNA-seq). The main innovation is the use of transient perturbation methods: modified mRNA for MYOD1 overexpression and siRNA for PRRX1 suppression instead of permanent genome-editing approaches such as CRISPR or lentiviral delivery. The dataset contains approximately 20,000 fibroblast cells across four experimental conditions (MYOD1 activation, PRRX1 suppression, dual perturbation, and control). Imaging was captured every 20 minutes for 72 hours using FUCCI cell cycle reporters. Benchmarking with Geneformer (foundation model), GEARS (perturbation prediction), and FM2S (image denoising) demonstrates high concordance between predicted and observed perturbation effects (cosine similarities 0.95–0.99).

**Strengths:**

The experimental methodology is rigorous, with well-chosen cell lines (BJ fibroblasts with FUCCI reporters), careful perturbation timing with serum starvation synchronization, and appropriate sequencing and imaging protocols. However, limitations reduce the depth of evaluation. The benchmarking strategy, especially Geneformer’s in-silico perturbation by simply setting MYOD1 to max and PRRX1 to zero is simplistic and does not reflect realistic transient kinetics of mmRNA/siRNA delivery. The study lacks technical replicates, validation experiments, and statistical analyses of differential expression or cell-cycle-specific effects.

The paper is clearly written and well organized. The authors describe data formats and public release clearly.

HYPED is among the first multimodal datasets combining transient RNA-based perturbations with long-read single-cell sequencing and time-resolved imaging, addressing a key gap in clinically relevant perturbation modeling. The dataset will be valuable for benchmarking multimodal and temporal machine learning approaches. The contribution is primarily experimental: no new algorithms or analytical frameworks are introduced, and the study covers only three perturbation conditions on a single cell type with what appears to be one biological replicate. The paper’s impact would increase substantially with deeper biological validation, expanded perturbation space, and additional sequencing timepoints.

●	Transient mmRNA/siRNA perturbations avoid genomic integration, aligning with therapeutic use cases.

●	Concurrent high-temporal-resolution imaging and transcriptomics capture both phenotypic and molecular dynamics.

●	FUCCI reporters provide fine-grained temporal context often ignored in reprogramming datasets.

●	Oxford Nanopore technology offers isoform-level and alternative-splicing information absent from short-read methods.

●	Data, preprocessing, and benchmarking code are publicly available.

●	Inclusion of Geneformer, GEARS, and FM2S establishes initial performance metrics for future work.

**Weaknesses:**

●	Only three perturbations on one cell type restrict generalizability; no biological replicates are reported.

●	In-silico perturbations and limited gene coverage (GEARS top 500 genes) may fail to capture biological complexity.

●	Lacks differential expression, pathway enrichment, and trajectory analyses confirming that perturbations drive myogenic fate (e.g., upregulation of MYOG, MYH3).

●	It is not explicit whether imaging and sequencing correspond to the same cells or just the same population.

●	No transfection efficiency, hashing accuracy, or batch-effect analysis provided.

**Questions:**

1.	Are imaging and sequencing performed on the same individual cells or on separate but matched populations?
2.	Do perturbed cells show induction of canonical myogenic markers, validating successful reprogramming?
3.	What are the actual timecourses of MYOD1/PRRX1 expression post-transfection?
4.	What are demultiplexing accuracy, QC pass rates, and transfection efficiencies?
5.	Why evaluate GEARS on only the top 500 genes, and how does it compare to other perturbation-prediction methods?

Show differential expression and pathway enrichment confirming myogenic shift; report transfection/QC statistics; provide graded rather than binary in-silico perturbations for Geneformer; and evaluate at least one temporal-prediction baseline using imaging data.

**Details Of Ethics Concerns:**

an explicit statement of cell-line sourcing and institutional approvals would improve clarity.

---

> ### Author Response · Authors · 2025-11-22
>
> We thank Reviewer G18c for their careful review and comments. Sequencing data had 3 replicates (separate wells) but were pooled together for single-cell barcoding and sequencing. For the Imaging data, the data contains 2 or 3 replicates each. The RT-qPCR and Western blot data, as well as other requested analyses are being prepared for publication in a separate manuscript and were not included in the present manuscript. A baseline method for temporal alignment of the two modalities has been included in the codebase and will be released as a part of the final submission.
>
> &nbsp;
>
> > **Are imaging and sequencing performed on the same individual cells or on separate but matched populations?**
>
> Imaging and sequencing were not performed on the same individual cells. Simultaneous capture of multiple modalities from a single cell remains a technical challenge and controlled measurements of matched populations is the trusted method for multimodal biological data acquisition. The cells captured on HYPED belong to the same cell type, cell cycle synchronized by serum starvation and treated under identical conditions, while capturing a novel combination of modalities.
>
> &nbsp;
>
> > **Do perturbed cells show induction of canonical myogenic markers, validating successful reprogramming?**
>
> The perturbed cells show upregulation for several myogenic genes, as well as downregulation of the fibroblast program. However, since MYOD1 was overexpressed for only 24 hours, we do not expect the cells to be fully reprogrammed, and canonical markers such as MYOG and MYH3 are not yet expressed. According to other datasets of MyoD-induced transdifferentiation [1], these downstream MYOD1 targets begin to increase in expression after approximately 72 hours.
>
> [1] Cacchiarelli et al. (2018) “Aligning Single-Cell Developmental and Reprogramming Trajectories Identifies Molecular Determinants of Myogenic Reprogramming Outcome.” Cell Systems, 7(3), P258-268.E3
>
> &nbsp;
>
> > **What are the actual timecourses of MYOD1/PRRX1 expression post-transfection?**
>
> We have RT-qPCR and Western blot data for 0, 24, 48, and 72 hours post-transfection. We have verified PRRX1 gene suppression 24-72 hours post-transfection and have confirmed MYOD1 gene overexpression at the scRNA-seq timepoint (24h) with this data. To provide an estimate, we expect MYOD1 protein to be overexpressed for at least 50-72 hours post-transfection [1, 2]
>
> [1] Karikó et al. (2008) “Incorporation of Pseudouridine Into mRNA Yields Superior Nonimmunogenic Vector With Increased Translational Capacity and Biological Stability.” Molecular Therapy, 16(11), 1833-1840
>
> [2]  Ferizi et al. (2015)  “Stability analysis of chemically modified mRNA using micropattern-based single-cell arrays.” Lab Chip, 15, 3561-3571
>
>
> &nbsp;
> > **What are demultiplexing accuracy, QC pass rates, and transfection efficiencies?**
>
> 1. For both scRNA-seq libraries (Hybrid + Control), demultiplexing was performed through the EPI2ME wf-single-cell pipeline. Barcode assignment statistics have been attached separately
>
> 2. Low quality cells were identified using a median absolute deviation (MAD)-based approach applied to each dataset. For each dataset, cells were flagged as outliers if their log-transformed total counts or number of detected genes were >5 MADs from the group median. Mitochondrial outliers were defined as cells with % mitochondrial counts >20% or >5 MADs from the group median. 87% of Hybrid cells and 78% of Control cells were of sufficient quality. Additionally, low quality genes were identified as those with less than 10 counts, with 87.3% of genes passing this QC metric.
>
> 3. Prior work in our lab using the same lipid-nanoparticle delivery system with mCherry mmRNA demonstrated a transfection efficiency of 77.7% (measured as mCherry-positive cells). In the present study, transcriptional changes support successful induction of both perturbations. Specifically, while 0% of control cells expressed MYOD1, 82.9% of mmMYOD1-treated cells showed detectable MYOD1 expression. For PRRX1 knockdown, 30.6% of control cells did not express PRRX1 compared to 66.7% of siPRRX1-treated cells, and 86.2% of siPRRX1-treated cells exhibited PRRX1 levels below the median PRRX1 expression observed in controls. Together, these data indicate robust uptake and activity of both mmRNA delivery and siRNA-mediated suppression.
>
> &nbsp;
> > **Why evaluate GEARS on only the top 500 genes, and how does it compare to other perturbation-prediction methods?**
>
> The GEARS model was fine-tuned with control data from HYPED for baseline expression with the 5000 most highly variable genes. Inference was performed on the three conditions and the predicted gene expression data was compared with the experimentally perturbed data. For this specific perturbation experiment, the top 500 genes should capture most of the variation in expression patterns. For reference, the authors for GEARS use only the top 20 genes as a comparison for their double gene perturbations.

---

> > ### Author Response · Authors · 2025-11-22
> > **Barcode assignment statistics**
> >
> > | Demultiplexing Stats                           | Hybrid        | Control       |
> > |------------------------------------------------|---------------|---------------|
> > | Total reads evaluated for barcode assignment   | 182,772,743   | 156,280,367   |
> > | Exact barcode match (%)                        | 76.65%        | 74.49%        |
> > | Corrected barcode match (%)                    | 8.79%         | 8.52%         |
> > | Multiple barcode hits (%)                      | 2.60%         | 2.18%         |
> > | No barcode hits (%)                            | 11.96%        | 14.80%        |
> > | Valid cell barcodes                            | 10,895        | 8,963         |
> > | High quality reads                             | 134,381,035   | 102,222,758   |
> > | High quality reads assigned to valid barcode   | 126,307,883   | 93,257,239    |
> > | Demultiplexing accuracy                        | 94%           | 91%           |

---

> > > ### Comment · Reviewer_G18c · 2025-11-28
> > >
> > > Thanks very much for the authors' clarification.
> > >
> > > It'll be better to expand table 1 to include more recent datasets in this field, e.g. Cell Painting series [1], the NeurIPS 2024 competition[2], etc.
> > >
> > > I still have some concerns on the quality of the dataset until the dataset is released. But this is a good step. I'll look forward to that.
> > >
> > > Reference
> > >
> > > [1] https://www.nature.com/articles/s41592-024-02241-6
> > > [2] https://openreview.net/forum?id=WTI4RJYSVm&referrer=%5Bthe%20profile%20of%20Mengbo%20Wang%5D(%2Fprofile%3Fid%3D~Mengbo_Wang1)#discussion

---

### Official Review · Reviewer_kYJp · 2025-10-31

**Soundness:** 3
**Presentation:** 3
**Contribution:** 3
**Rating:** 6
**Confidence:** 2

**Summary:**

The paper introduces HYPED a dataset for studying how human fibroblasts respond to transcription factor perturbations using RNA-based delivery methods instead of the permanent genetic modifications used in most already existing datasets. The key contribution is combining two data types, long-read single-cell RNA sequencing and time-lapse microscopy images taken every 20 minutes over 72 hours with cell cycle markers. The author argue this is important because transient RNA-based perturbations are more clinically relevant than CRISPR approaches, and the temporal imaging captures dynamics that are usually missed. They benchmark a few existing models on each modality.

**Strengths:**

* As far as I can tell, this is genuinely the first perturbation dataset using transient RNA-based methods (mmRNA/siRNA) combined with multimodal measurements. Most existing datasets use CRISPR or viral integration which permanently alters DNA, and the authors make a convincing case for why this matters clinically and scientifically.
* The long-read sequencing is a nice touch. Being able to capture full-length transcripts and isoform diversity goes beyond what standard short-read scRNA-seq provides. Though I'll admit I'm not expert enough in sequencing to fully evaluate how much this matters in practice for the perturbation prediction problem.
* The data documentation and accessibility seem solid. This suggest they've thought carefully about making this usable for ML researchers.

**Weaknesses:**

* My biggest concern is that this feels more like a dataset paper that stops short of demonstrating what makes it valuable for ML. They benchmark Geneformer and GEARS on sequencing and FM2S on imaging, but these are all unimodal evaluations. The whole point of collecting both modalities should be to enable multimodal learning, but there's no demonstration of this. What does jointly modeling imaging and transcriptomics buy you? Can you predict sequencing from imaging or vice versa? Can multimodal models better predict perturbation outcomes? Without answering these questions, I'm left wondering if the multimodal aspect is actually useful.
* The scope is quite limited, only three perturbation conditions (plus control) on one cell type. I understand this is acknowledged in the limitations, but it raises questions about whether this dataset is substantial enough for training modern foundation models or even properly evaluating perturbation prediction methods.
* The imaging is continuous over 72 hours, but sequencing happens at a single endpoint (48h). How do you connect individual cells imaging trajectories to their transcriptomic states? You can't directly match them since sequencing destroys cells. The paper mentions collecting from "the same population" but doesn't explain how you'd actually use both modalities together for a single cell. This seems like a fundamental challenge for multimodal modeling that's just glossed over.

**Questions:**

* Why are the cosine similarities so high (>0.95) in the benchmarking? Does this suggest the perturbations are subtle, or that current models are already quite good at this task, or something about the evaluation setup?
* Can you clarify the temporal alignment between modalities? Since sequencing requires lysing cells while imaging is non-destructive, how would a multimodal model actually use both data types for the same cell? Is the idea that you learn population-level associations, or is there a way to track specific cells?
* What specific multimodal learning tasks do you envision for this dataset? Can you predict imaging phenotypes from transcriptomics or vice versa? Would jointly modeling both modalities improve perturbation prediction compared to using either alone?

---

> ### Author Response · Authors · 2025-11-22
>
> We thank Reviewer kYJp for their thoughtful and detailed comments. To the reviewer’s second point, we believe that the choice of 3 perturbation types does not necessarily limit generalizability. During reprogramming only a small percentage of the source cells reach the target state due to phenotypic variations arising from cell dynamics. The heterogeneous nature of cell states can be captured by single cell RNA sequencing while the dynamics can be captured by the imaging modality providing sufficient conditions for the model to generalize. The reviewer’s questions have been addressed in depth below
> &nbsp;
> > **Why are the cosine similarities so high (>0.95) in the benchmarking? Does this suggest the perturbations are subtle, or that current models are already quite good at this task, or something about the evaluation setup?**
>
> From the benchmarking performed, we infer that existing models reflect real world behavior accurately. The perturbation prediction models are accurate and the experimental measurements serve as empirical ground truth for change in gene expression in reprogramming. We believe that this positions HYPED dataset as a reliable ground truth for future models trained jointly on both imaging and sequencing modalities.
>
> &nbsp;
> > **Can you clarify the temporal alignment between modalities? Since sequencing requires lysing cells while imaging is non-destructive, how would a multimodal model actually use both data types for the same cell? Is the idea that you learn population-level associations, or is there a way to track specific cells?**
>
> The two modalities captured as part of HYPED include live-cell Imaging and gene-expression. These modalities were not performed on the same individual cells, the population of cells used for sequencing is different from the population of cells for which time series cell cycle imaging data was captured. Capturing multiple modalities from a single cell still remains a technical challenge and measurements of matched populations is the trusted method for multimodal biological data acquisition. These cells are of the same cell type and were cell cycle synchronized by serum starvation for 24 hours.
>
> Our proposed method for aligning cells between modalities is to infer the pseudotemporal ordering of cells [1] from gene-expression data and coupling this with the expressed phenotypic characteristics from a relative time point. Even though the exactly matched characteristics from both modalities are not captured, our dataset provides a good starting point for coupling these two modalities that have only been analyzed separately.
>
> [1] Haghverdi, Laleh, et al. "Diffusion pseudotime robustly reconstructs lineage branching." Nature methods 13.10 (2016): 845-848
>
> &nbsp;
> > **What specific multimodal learning tasks do you envision for this dataset? Can you predict imaging phenotypes from transcriptomics or vice versa? Would jointly modeling both modalities improve perturbation prediction compared to using either alone?**
>
> The main multimodal learning task we envision for our dataset is predicting the gene expression from live cell imaging data. Modeling and incorporating captured changes in cell cycle and morphology of the cells will improve future perturbation prediction models, as current perturbation prediction benchmarks do not take these Phenotypic responses into consideration. We envision that HYPED will enable training and validation of models capable of inferring time-series gene expression dynamics from imaging data.

---

### Official Review · Reviewer_i3fi · 2025-10-31

**Soundness:** 4
**Presentation:** 4
**Contribution:** 4
**Rating:** 8
**Confidence:** 4

**Summary:**

This paper describes a new dataset that can be used to train ML models to predict the effect of transient perturbation of transcription factors.  The data contains unpaired long-read scRNA-seq and imaging data.  The paper does a good job of explaining the utility of such data, and how it relates to and improves upon existing benchmark datasets.

**Strengths:**

This is clearly a valuable and well designed dataset that will be of broad interest to people developing models related to cellular reprogramming.  The manuscript is very clearly written, and the benchmarking provided here gives some initial insights into the types of analysis that can be carried out with this data.

**Weaknesses:**

I found it strange that there was no discussion of the results, after section 4 and before the discussion.

Line 53: Add comma after "DNA."

Line 86: I don't actually understand this sentence.  Why does changing the DNA necessarily imply that the cell cycle is not taken into account?  Similarly, maybe existing ML models don't take into account cell cycle, but I don't see why they couldn't try to do so.

Where does the name "HYPED" come from?  Seems like it should be explained somewhere.

Table 1 is very helpful in making clear the contribution here.

I was't quite clear on why you talk about FUCCI (line 134).  It seems like you use a different set of cell cycle markers (line 324).  This should be clarified in the paragraph at line 134.

Line 163: "of the same cells"  I think you should clarify here that the dataset is unpaired; i.e., you don't know which image corresponds to which scRNA-seq profile.

Line 249: You should indicate the size of the dataset (number of cells and of images) in the abstract and introduction.

Line 323: Misspelling ("delineate")

Line 329: Tense shift in this paragraph.

Line 345: Cosine -> cosine

Line 351: Add a cite for rank value encoding.

Line 359: Give details of how the train/test/validation splits are done.  Store the splits info online somewhere public.  And don't capitalize "Test" and "Train."

Line 366: "image processing techniques" is too vague.  Give details, either in the manuscript or in a supplement.

Line 369: Missing word before "various."

Line 471: Anonymization problem here.

**Questions:**

Is FUCCI used in HYPED?

Why does changing the DNA necessarily imply that the cell cycle is not taken into account?

What are the conclusions that you draw from Table 3 and Table 4?

---

> ### Author Response · Authors · 2025-11-22
>
> We thank the reviewer i3fi for their thoughtful review of our manuscript. The suggestions make the manuscript much stronger and have been incorporated into the updated version.
> &nbsp;
> > **Is FUCCI used in HYPED?**
>
> Yes, FUCCI is used in HYPED. The underlying mechanism of the Incucyte Cell Cycle Reporter (Sartorius 4779) used  in HYPED is the Fluorescent ubiquitination-based cell cycle indicator (FUCCI) system. This captures cell-cycle dependent changes in the expression patterns of Geminin and Cdt1 by linking fluorescent proteins TagGFP2 (Green) and mKate2 (Red) that can be monitored with imaging.
> &nbsp;
> > **Why does changing the DNA necessarily imply that the cell cycle is not taken into account?**
>
> Changing the DNA does not imply that the cell cycle is not taken into account. The main focus of line 086 was to show that perturb-Seq and current machine learning benchmarks don’t take the cell-cycle dynamics into account. The corresponding line has been modified and positioned appropriately in the updated manuscript.
> &nbsp;
> > **What are the conclusions that you draw from Table 3 and Table 4?**
>
> From the benchmark Table 3, we observe high cosine similarities between the predicted expression and experimentally captured expression across all three perturbation conditions. The Geneformer benchmark shows that the gene representations learned by the model align with experimental real world measurements. The trained machine learning models accurately reflect experimental behavior and our dataset serves as a strong empirical ground truth for training and evaluating future foundation models that can leverage both modalities captured in HYPED. From Table 4, we can evaluate the quality of segmentation masks generated. Out of the box noise removal method FM2S was able to remove noise efficiently on live cell imaging data.

---

### Note · Program_Chairs · 2026-01-17
**Submission Desk Rejected by Program Chairs**

The following references in this submission do not refer to real documents and/or have major errors in bibliographic information:

 Anna K Blakney, Paul F McKay, and Robin J Shattock. Delivery of mrna-based vaccines and therapeutics with lipid nanoparticles. Biochemical Society Transactions, 47(4):1209-1218, 2019.